# Response of Sap Flow Trends of Conifer and Broad-Leaved Trees to Rainfall Types in Sub-Humid Climate Region of China

Yongxiang Cao [1,2], Yushi Wang [3], Naichang Zhang [1], Chendong Ning [1], Yu Bai [1] and Jianbo Jia [4,*]

[1] Power China Northwest Engineering Corporation Limited, Xi'an 710065, China; caoyongx@nwh.cn (Y.C.); zhangnc111123@163.com (N.Z.); ningcd123124@163.com (C.N.); jotham880303@126.com (Y.B.)
[2] Shaanxi Union Research Center of University and Enterprise for River and Lake Ecosystems Protection and Restoration, Xi'an 710065, China
[3] Beijing Forestry University, Beijing 100083, China; wangys123124@163.com
[4] Central South University of Forestry & Technology, Changsha 410004, China
* Correspondence: jiajianbo@csuft.edu.cn

**Abstract:** Sap flow is one of the most important physiological water transport processes of trees, and the characteristics of sap flow are greatly affected by the spatial and temporal distribution of water in the SPAC (soil–plant–atmosphere continuum). However, different precipitation characteristics have great influence on the water environment of forest trees, which causes considerable differences in sap flow. Therefore, researching the response of sap flow to precipitation type is the key to accurately determining plant transpiration in semi-arid areas. We used K-means clustering analysis to divide the rainfall during the study period into three rainfall types (the highest rainfall amount and intensity (types I), medium rainfall amount and intensity, with a long duration (types II); and the lowest rainfall amount and intensity (types III)) based on the rainfall amount and intensity in order to compare the differences in the response of sap flow trends and influencing factors of *Pinus tabulaeformis* and *Robinia pseudoacacia* under different rainfall types. The results showed that, under the daily scale average sap flow of *P. tabulaeformis* and *R. pseudoacacia*, rainfall type II decreased significantly relatively to rainfall types I and III ($p < 0.05$). In rainfall type II, The sap flow characteristics of *R. pseudoacacia* were positively correlated with solar radiation ($p < 0.05$), while those of *P. tabulaeformis* were positively correlated with temperature, solar radiation, and VPD ($p < 0.01$). The sap flow of *P. tabulaeformis* and *R. pseudoacacia* were significantly positively correlated with temperature, solar radiation, VPD, and soil moisture content ($p < 0.01$) and negatively correlated with relative humidity ($p < 0.05$) in rainfall type III. The hourly sap flow of *P. tabulaeformis* and *R. pseudoacacia* on rainfall days was higher than before the rainfall. Rainfall type I promoted the daily sap flow of both species, and the proportion of the sap flow in daytime was also higher. On rainy days, the sap flow rates of rainfall type I and III showed a "midday depression". In type I rainfall events, the sap flow "midday depression" after rainfall occurred an hour ahead compared to the sap flow "midday depression" before rainfall. In type II rainfall events, the daytime sap flow rates of *P. tabulaeformis* and *R. pseudoacacia* were obviously inhibited, but the nighttime sap flow rate increased. In type III rainfall events, the sap flow before rainfall presented a unimodal curve versus time. The daily average sap flow of *R. pseudoacacia* was more susceptible to rainfall type II, while *P. tabulaeformis* was more susceptible to rainfall types I and III. The sap flow rate of *R. pseudoacacia* decreased on rainy days. The results show that the effects of different rainfall types on the sap flow trends of *P. tabulaeformis* and *R. pseudoacacia* were different. They revealed the responses of their sap flow trends to meteorological factors under different rainfall types, which provided basic data and theoretical support for further predicting the sap flow trends on rainy days, clarifying the effects of rainfall amount, rainfall duration, and rainfall intensity on sap flow trends and accurately estimating the transpiration water consumption of typical tree species in the sub-humid climate regions of China.

**Keywords:** sap flow; rainfall types; K-means clustering; sub-humid climate

## 1. Introduction

Water plays an important role in vegetation distribution and plant life activities [1]. Precipitation, as a crucial element of climate conditions, is straightforwardly linked to the growth of plants, biodiversity, and the stability of forest ecosystems, fundamentally impacting almost all hydrological processes within the forest ecosystem [2–4]. Under the action of transpiration pull, plants absorb moisture in the soil through roots, transport it to various parts through ducts, and output it into the atmosphere through the transpiration of the leaves [5,6]. Sap flow is not only one of the most important physiological processes of trees, but it also reflects the plant's water transport capacity and sensitivity to environmental changes [7,8]. At the same time, sap flow is a key link in the SPAC (soil–plant–atmosphere continuum), which transports the soil water absorbed in roots and determines the amount of tree transpiration [9–11]. This can be used to analyze the characteristics of water consumption and the water transport mechanism of trees [12,13]. Climate change affects global hydrological processes and changes the spatial and temporal patterns of continental precipitation [14–16]. The most important characteristics are the decrease in precipitation frequency, the uneven spatial distribution of precipitation, the increase in extreme precipitation, and the increase in seasonal drought in sub-humid climate regions [8,17]. Precipitation is an important supplement to soil water, and changing temporal and spatial precipitation patterns also affect plant water use and canopy transpiration [18,19]. The extent to which vegetation is affected by rainfall patterns is related to different strategies through which tree species deal with drought [20,21]. Therefore, it is of great importance to research the correlation between precipitation patterns and soil water use of trees in seasonal arid areas under climate change [22,23].

The sap flow characteristics of different tree species are greatly different under the influence of physiological structures, water conditions, and meteorological factors [24,25]. In the long term, sap flow is mainly affected by root-related factors, while in the short term, it is more sensitive to meteorological factors [26–28]. Therefore, the time-delay effect of environmental factors can be used to simulate sap flow characteristics [29]. Previous studies have shown that the characteristics of sap flow show different variation patterns under the influence of weather [30–32]. On sunny days, the sap flow tends to be high in the daytime and low at night, and the variation trend is a single-peak curve or double-peak curve [8]. On rainy days, on the other hand, the curves fluctuate greatly or show gentle trends [33]. Therefore, the sap flow is one of the most important physiological water transport process of trees, and the characteristics of sap flow are greatly affected by the water environment [34,35]. However, different precipitation characteristics have a great influence on the water environment of forest trees, which leads to significant differences in sap flow. The response of sap flow to rainfall types is the key to accurately calculating plant transpiration [36,37].

In order to study the transpiration characteristics of trees, most studies on the relationship between sap flow and environmental factors were mainly conducted on typical sunny days [34,38]. Other studies have found that other rainfall parameters, such as rainfall duration and rainfall intensity, also affect plant transpiration in dryland ecosystems [35]. However, there have been few studies on the response of the sap flow characteristics of trees to rainfall types. The main objectives of this study were: (i) to explore the differences in the response of sap flow characteristics to different rainfall types and (ii) analyzing the difference in sap flow between conifers and broad-leaf trees under different rainfall types, as well as its influencing factors. Our purpose was to provide data supporting the study of vegetation transpiration and hydrological process. Our study aimed to investigate the effects of different rainfall events on sap flow for the purpose of providing water use efficiency, which is a supplement to the research on vegetation transpiration and hydrological processes.

## 2. Materials and Methods

### 2.1. Study Site Description

This study was performed in Jiu Feng National Forestry Park, Haidian District, about 30 km northwest of Beijing, China (39°540 N, 116°280 E) (Figure 1). It is a semi-humid region situated 140 m above the mean sea level, with a 10–25° sloping terrain. The mean daily temperatures range between 23 and 28 °C from May to mid-October, and between −5 and +25 °C throughout the winter, with a mean annual temperature of 12 °C. The mean annual precipitation is 630 mm, and the maximum rainfall occurs between June and September. In the study area, *P. tabulaeformis* and *R. pseudoacacia* are the dominant conifer and broad-leaved tree species. Most of these species were planted in the 1950s and 1960s. Lolium perenne, the main herbaceous vegetation (mean 70% ground cover), is widely distributed in the study area. The main shrubs were *Grewia biloba*, *Vitex negundo*, and *Broussonetia papyrifera*, whose densities were 3–5 plants/m². The soil in the study area was cinnamon soil, with an average slope of approximately 10% in the northeastern direction.

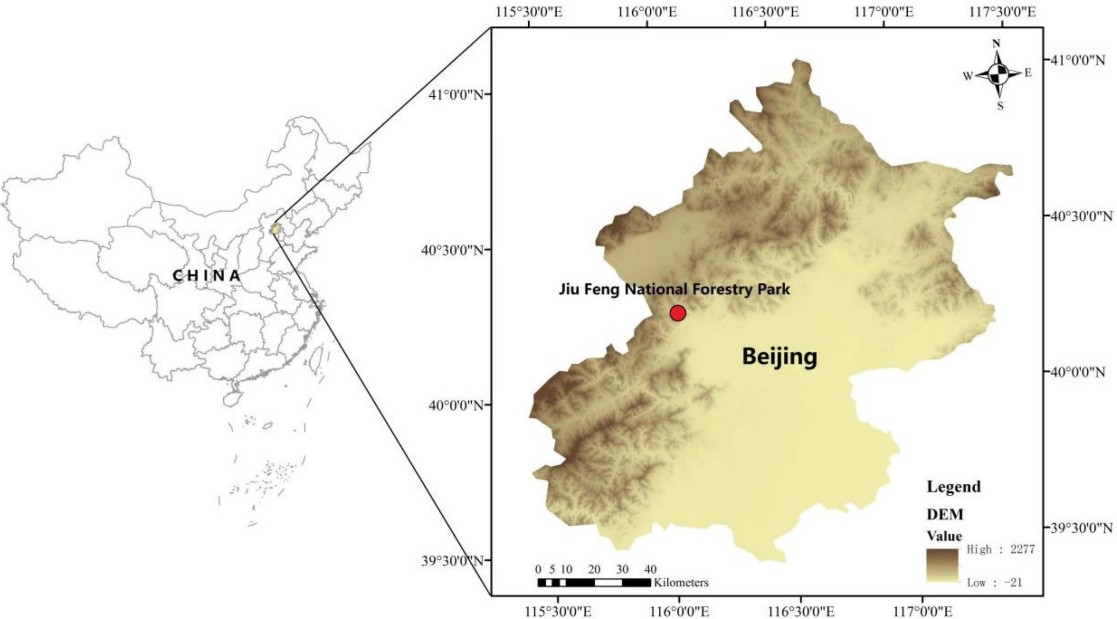

**Figure 1.** The location of Jiu Feng National Forestry Park in Beijing, China.

Three fixed observation stands (20 m × 20 m) of *P. tabulaeformis* and three fixed observation stands (20 m × 20 m) of *R. pseudoacacia* were selected and numbered. According to the average DBH and height, three standard sample trees with good growth and straight trunks without disease or insect pests were selected in each observation stand. The conditions of the selected plots were similar, e.g., in terms of altitude, slope, and aspect, which was advantageous for comparing the difference in sap flow between the two species under the same environmental conditions. The basic information on the plots and trees is shown in Table 1.

**Table 1.** Information of observation stands and average values of sample trees.

| Species Stand | Density/(Tree/hm²) | Average Age/a | Indices | Standard Sample Trees | | |
| | | | | No.1 | No.2 | No.3 |
| --- | --- | --- | --- | --- | --- | --- |
| *P. tabuliformis stand* | 1200 | 25 | Height/m | 8.5 | 5.3 | 6.2 |
| | | | diameter/cm | 18.47 | 16.57 | 13.06 |
| *R. pseudoacacia stand* | 900 | 43 | Height/m | 9.6 | 9.1 | 9.4 |
| | | | diameter/cm | 23.89 | 17.83 | 14.65 |

### 2.2. Sap Flow Measurements

A thermal dispersive sap flow measurement system (FLGS-TDP XM1000, Dynamax, Houston, USA) was used for continuous monitoring (the monitoring period lasted from April 2020 to September 2021). As for the calibration of Granier sensors, we have already considered radial calibration and tested the accuracy of the flow when installing the Granier sensors. Two kinds of probes (20 mm and 30 mm) were selected according to the xylem catheter in the radial position to ensure the accurate position of the probe. The collection frequency was set to 15 min to record once, and the sap flow rate was calculated according to the empirical formula proposed by Granier [38].

$$J_s = 0.0119 \times \left( \frac{d_{T_{max}} - d_T}{d_T} \right)^{1.231} \tag{1}$$

where $J_s$ is the sap flow rate (g·cm$^{-2}$·s$^{-1}$), $d_{Tmax}$ is the maximum temperature difference between the upper and lower probes within a day (°C), and $d_T$ is the instantaneous temperature difference at the time of measurement (°C).

The daily sap flow (g·d$^{-1}$) of the whole tree was calculated according to the average sap flow rate and the sapwood area (cm$^2$) of the sample tree, and the formula was as follows:

$$SF = J_s \times A_s \times 3600 \times 24 \tag{2}$$

Our research focuses on the influence of different rainfall events on sap flow, that is, the difference analysis of sap flow before rainfall and after rainfall. According to some rainfall events research literature [39,40], we defined the interval between two rainfall events as more than 24 h; the sap flow before rainfall (or pre-rainfall) refers to the sap flow within 24 h before a rainfall event begins, and the sap flow after rainfall (or rainfall) refers to the sap flow within 24 h before a rainfall event stops. The sapwood area was calculated according to sample trees with a similar DBH and height in the fixed observation stand, and wood cores were taken with growth cones to measure DBH and sapwood thickness to calculate the sapwood area. For basic information on the sample trees and sapwood area calculation, we refer the reader to our previously published literature [39].

### 2.3. Soil Water and Meteorological Factor Measurements

According to the previous studies on the distribution range of rhizosphere and the depth of root water absorption of two species of trees [40], we measured the average soil water content (SWC) of 0–40 cm soil layers using soil water sensors (ECH2O-TE System, Decagon Devices, Pullman, Washington, USA). At the same time, meteorological data, which included the characteristic values of precipitation (rainfall $R_a$, rainfall intensity $R_i$, rainfall duration $R_d$), net radiation ($R_n$), air temperature ($T_a$), relative humidity of the air (RH, %) and wind speed ($W_S$), were recorded synchronously by the HOBO automatic small weather station (U30, Onset, MA, USA). The data were recorded at 15 min intervals. The vapor pressure deficit (VPD) was calculated from the air temperature and relative humidity, and the formula is as follows:

$$VPD = 0.61078 \times e^{\frac{17.27 \times T_a}{T_a + 240.97}} \times (1 - RH) \tag{3}$$

### 2.4. Cluster Analysis of Rainfall Events

Compared with traditional rainfall classification, the clustering classification based on rainfall characteristic values can more accurately depict the rainfall process. The measurement period lasted 316 days (from April 21 to 28 September 2020 and from 4 April to 5 September 2021). Therefore, 82 rainfall events (growing seasons for 2020 and 2021) were characterized by three precipitation characteristic values, which included the rainfall amount ($R_a$(mm)), rainfall duration ($R_d$(h)), and rainfall intensity ($Ri$(mm·h$^{-1}$)), were evaluated by the K-means clustering analysis method (Table 2). The classification met the criteria for one-way analysis of variance (ANOVA) at the level of significance ($p < 0.05$),

and the three clusters kept the overall intra-group variance to a minimum. Type I had the highest rainfall amount and intensity and medium rainfall duration; type II had a medium rainfall amount and the lowest intensity, but the longest rainfall duration; and type III had the lowest rainfall amount and medium rainfall intensity, but the shortest rainfall duration.

**Table 2.** Statistical characteristics of different rainfall types.

| Rainfall Types | Frequency | Rainfall Characteristic | Mean ± Standard Deviation | Variation Coefficient (%) |
|---|---|---|---|---|
| Type I | 3 (3.67%) | Rainfall amount/mm | 77.60 ± 74.10 | 95.43 |
| | | Rainfall duration/h | 7.83 ± 6.77 | 86.37 |
| | | Rainfall intensity/mm h$^{-1}$ | 13.27 ± 6.84 | 51.56 |
| Type II | 28 (34.15%) | Rainfall amount/mm | 15.64 ± 14.55 | 93.03 |
| | | Rainfall duration/h | 13.40 ± 5.19 | 38.71 |
| | | Rainfall intensity/mm h$^{-1}$ | 1.27 ± 1.37 | 107.35 |
| Type III | 51 (62.20%) | Rainfall amount/mm | 2.37 ± 3.60 | 152.40 |
| | | Rainfall duration/h | 2.00 ± 2.09 | 104.25 |
| | | Rainfall intensity/mm h$^{-1}$ | 1.48 ± 1.88 | 126.60 |

*2.5. Data Analysis*

Origin Pro 18, R and Minitab 19 were used to calculate the mean, standard deviation, coefficient of variation, and other statistical data of various indicators under different rainfall types. ANOVA was used to evaluate the statistical differences of daily meteorological factors under different rainfall types and the statistical differences of sap flow before and during rainfall between different rainfall types. Pearson correlation analysis was used to test the correlation between sap flow and influencing factors.

**3. Results**

*3.1. Precipitation Characteristics and Meteorological Factors*

Comparing the amount of rainfall and the duration and intensity of the 82 rainfall events in the two growing seasons, as well as the classification results of the K-means clustering analysis, it can be seen that rainfall events of type I occurred in July, August, and September of the first growing season (Figure 2). The amount and intensity were higher than the other types, but the occurrence frequency was low (3 events, 3.67%). Types II and III appeared alternately in two growing seasons, and the frequency of type III (51, 62.2%) was higher than that of type II (28, 34.15%). Type II was characterized by the lowest rainfall intensity and longer rainfall duration (over 8 h), while type III had the shortest rainfall duration (under 5 h) along with the lowest rainfall amount (less than 6 mm).

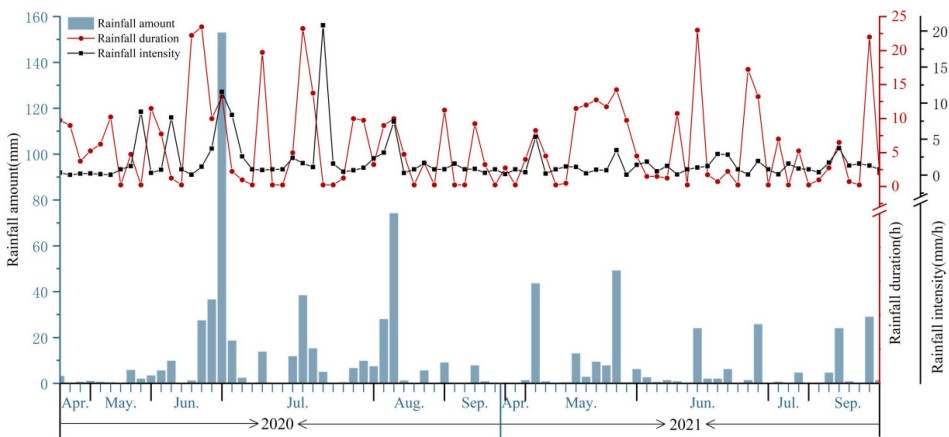

**Figure 2.** Precipitation characteristics of 82 rainfall events during two growing seasons.

The range of variation for $T_a$ (5.81–31.54 °C), Rn (0–494.36 W·m$^{-2}$), RH (7.13–100%), and VPD (0–4.30 kpa) during the two growing seasons showed obvious seasonal variations (Figure 3). The lowest values of monthly mean $T_a$ (13.28 °C), RH (38.06%), and VPD (0.99 kpa) appeared in April, and the highest values appeared in July and August (26.67 °C, 63.80%, 2.14 kpa). The lowest value of Rn appeared in September (94.03 W·m$^{-2}$) and the highest value occurred in May (145.40 W·m$^{-2}$).

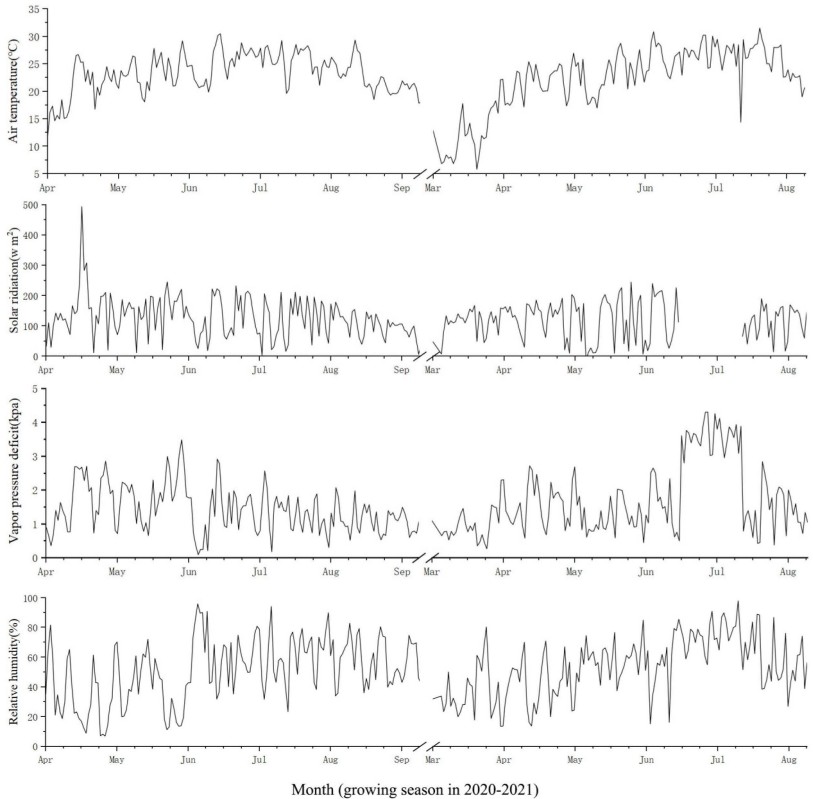

**Figure 3.** Meteorological trends during two growing seasons.

The daily mean meteorological factors of the three rainfall types are shown in Table 3 (Rn in July of the second growing season was missing due to equipment failure). The variation range of daily mean $T_a$, Rn, RH, and VPD on rainy days were 7.16–29.26 °C, 4.1–240.62 W·m$^{-2}$, 15.30–94.22%, and 0.18–2.65 kpa, respectively. The daily Rn of type I was significantly higher than that of type III ($p < 0.05$), but there were no significant differences in the $T_a$, Rn, or VPD of the three rainfall types.

**Table 3.** Characteristics of daily mean meteorological factors within different rainfall types during the growing seasons.

| Rainfall Types | $T_a$ (°C) | Rn (w·m$^{-2}$) | RH (%) | VPD (kpa) |
|---|---|---|---|---|
| Type I | 23.65 ± 3.76 a | 94.30 ± 106.80 a | 49.53 ± 15.91 a | 1.51 ± 0.20 a |
| Type II | 20.05 ± 4.49 a | 45.71 ± 41.31 ab | 50.96 ± 19.49 a | 1.19 ± 0.60 a |
| Type III | 22.20 ± 4.84 a | 95.31 ± 70.99 b | 56.65 ± 17.08 a | 1.17 ± 0.54 a |

Note(s): Different letters indicate significant differences in daily mean meteorological factors among the different rainfall types.

### 3.2. Daily Sap Flow Characteristics within Three Rainfall Types

The characteristics of sap flow under three rainfall types are shown in Figure 4. Type I only appeared in the growing season in 2020. Types II and III appeared to be evenly distributed in the two growing seasons, and the daily mean sap flow of *P. tabulaeformis* and *R. pseudoacacia* showed no obvious changes under the three rainfall types. The highest daily

sap flow values of *P. tabulaeformis* and *R. pseudoacacia* occurred in type I (25,986 g·d$^{-1}$ and 42,931 g·d$^{-1}$, respectively), and the lowest values occurred in the type II (6905 g·d$^{-1}$ and 14,656 g·d$^{-1}$, respectively). There were significant differences in the average daily sap flow of *P.tabulaeformis* before and after type II ($p < 0.05$), but no significant difference in the other two rainfall types.

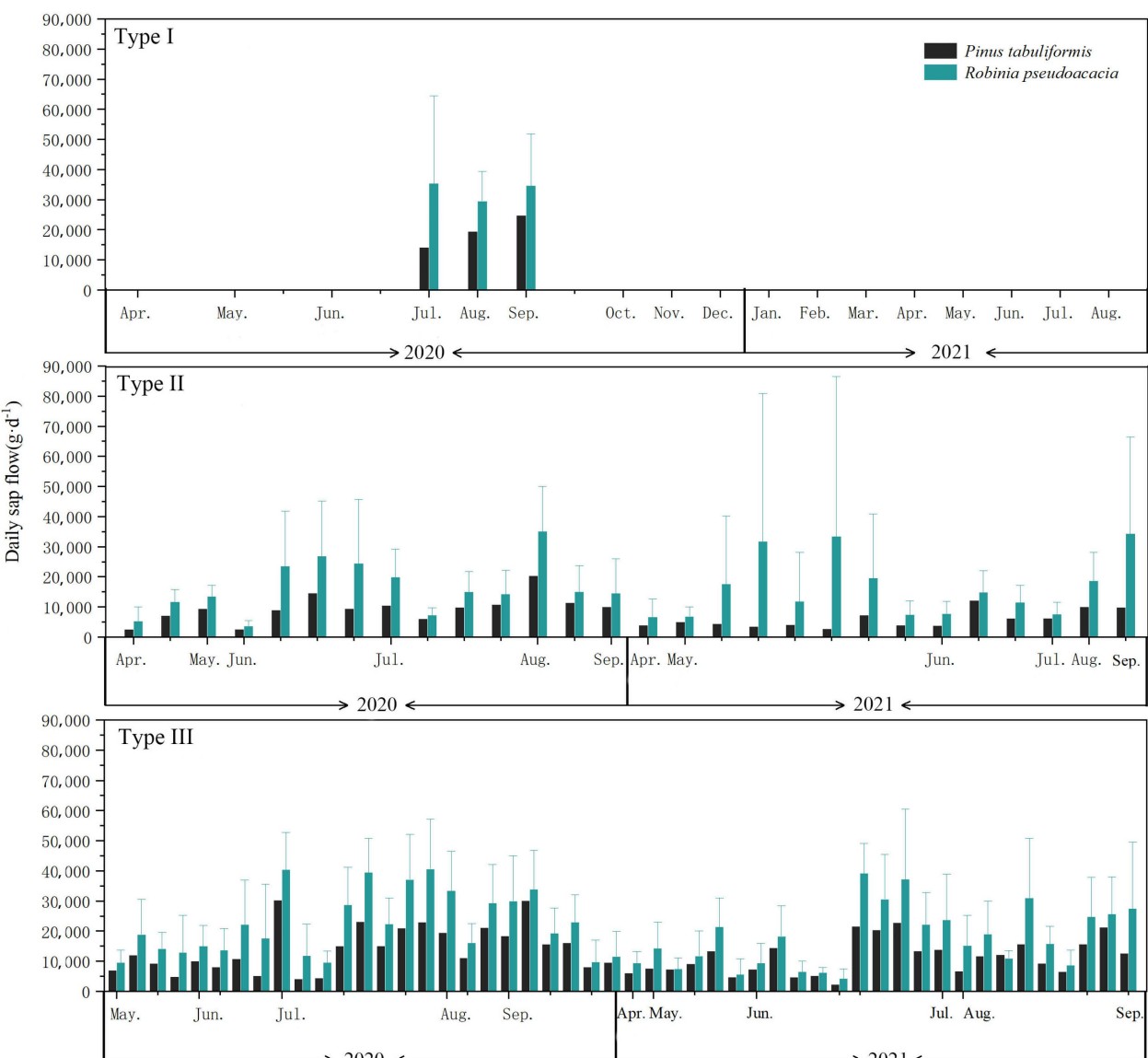

**Figure 4.** Daily sap flow characteristics of *P. tabuliformis* and *R. pseudoacacia* under three rainfall types in two growing seasons (the value was measured 24 h after rain).

Both before and after rainfall, the average daily sap flow of *P. tabulaeformis* in type II was significantly lower than that in types I and III ($p < 0.05$). Before rainfall, the average daily sap flow of *R. pseudoacacia* showed no significant differences among the three rainfall types. After rainfall, the average daily sap flow of *R. pseudoacacia* in type I was significantly higher than that in type II ($p < 0.05$), and that in type II was significantly higher than that in type III ($p < 0.05$). The daily sap flow of *R. pseudoacacia* was higher than that of *P. tabulaeformis* (Figure 5). Under type I, the daily values of *P. tabulaeformis* and *R. pseudoacacia* after rainfall increased by 12.6% and 9.3% relatively to before rainfall, and by 85.2% and 98.7% under type II. But under type III, the daily values decreased by 29.1% and 21.0%, respectively.

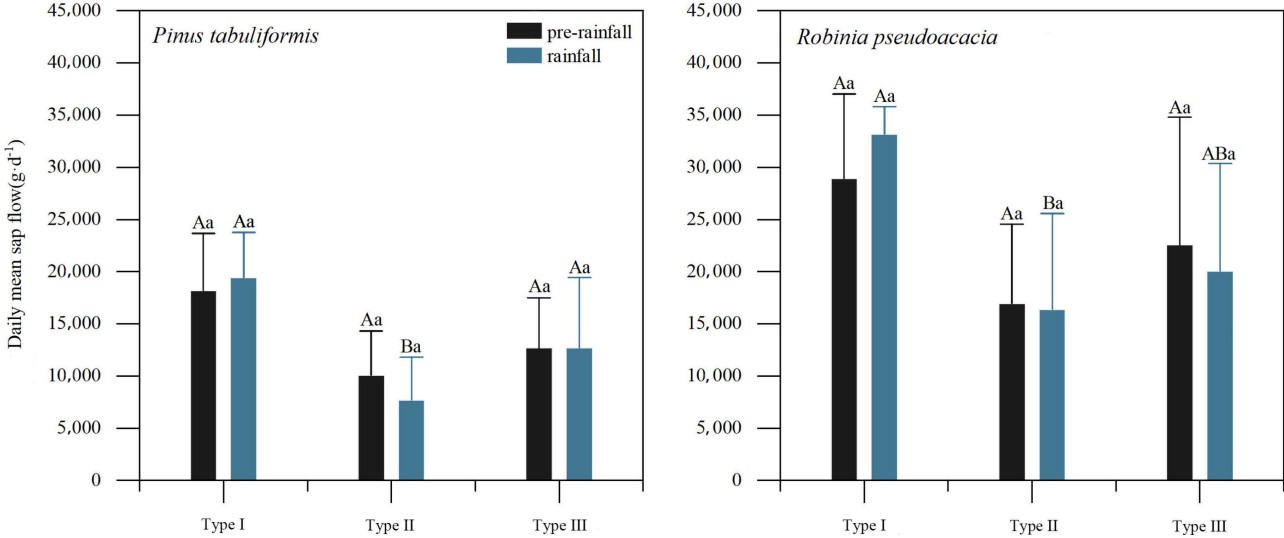

**Figure 5.** Comparison of daily pre-rainfall and after rainfall sap flow characteristics of *P. tabuliformis* and *R. pseudoacacia* under three rainfall types (different letters indicate significant differences in daily mean sap flow between the different rainfall types).

The correlation between the sap flows of *P. tabulaeformis* and *R. pseudoacacia* for types II and III, as well as the influencing factors, are analyzed in Table 4. The sap flow of *P. tabulaeformis* was positively correlated with temperature, soil water content, and solar radiation, and VPD ($p < 0.01$) for types II and III, and was negatively correlated with relative humidity for type III ($p < 0.01$). The correlation between the sap flow of *R. pseudoacacia* and the influencing factors was not consistent between the two rainfall types. Under type II, the daily sap flow of *R. pseudoacacia* was only positively correlated with solar radiation ($p < 0.05$), while under type III, the daily sap flow of *R. pseudoacacia* was positively correlated with temperature, solar radiation, VPD, and soil moisture content ($p < 0.01$). It was negatively correlated with relative humidity ($p < 0.05$).

**Table 4.** Correlation coefficients between daily sap flow and influencing factors of *P. tabuliformis* and *R. pseudoacacia* with rainfall types II and III.

| Species | Rainfall Types | T | SR | RH | VPD | SWC |
|---|---|---|---|---|---|---|
| *P.tabuliformis* | Type II | 0.60 ** | 0.73 ** | −0.25 | 0.52 ** | 0.31 |
| | Type III | 0.37 ** | 0.68 ** | −0.41 ** | 0.63 ** | 0.52 ** |
| *R.pseudoacacia* | Type II | 0.37 | 0.46 * | 0.17 | 0.09 | 0.15 |
| | Type III | 0.50 ** | 0.72 ** | −0.34 * | 0.66 ** | 0.62 ** |

Note: * Correlation is significant at the 0.05 level (2-tailed); ** Correlation is significant at the 0.05 level (2-tailed).

The linear relationship between the daily sap flows of *P. tabulaeformis* and *R. pseudoacacia* and the influencing factors varied for different rainfall types, as shown in Figure 6. The daily sap flows of *P. tabulaeformis* and *R. pseudoacacia* increased with the increase in solar radiation under the two rainfall types. Under type II, there was no obvious correlation between the daily sap flow of *R. pseudoacacia* and temperature or VPD, but under type III, with the increase in temperature, solar radiation, VPD, and soil water content, the increase in the daily sap flow of *R. pseudoacacia* was higher than that of *P. tabulaeformis*, and with the increase in relative humidity, the decrease in the daily sap flow of *R. pseudoacacia* was also higher than that of *P. tabulaeformis*.

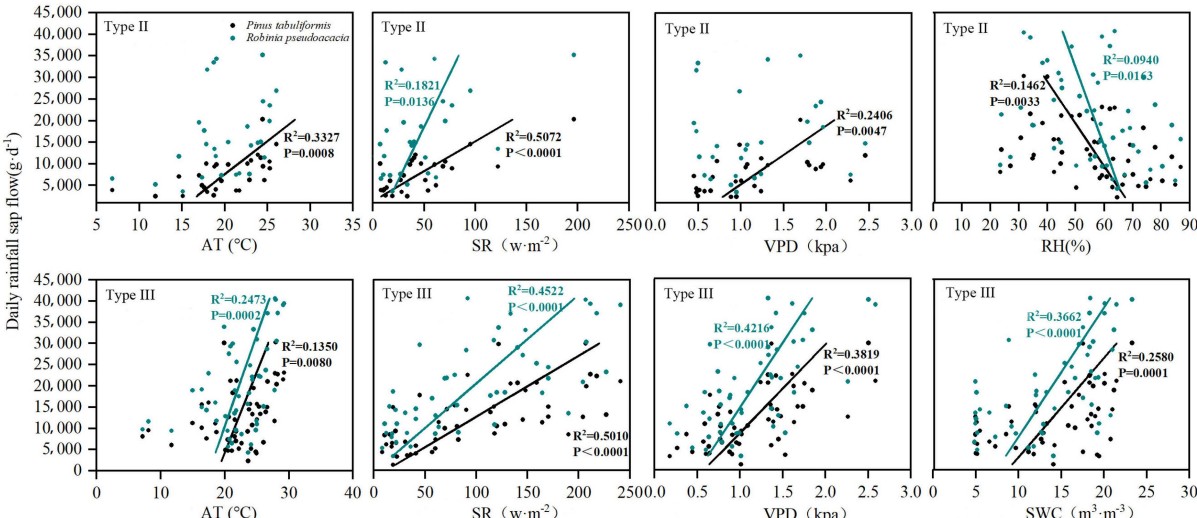

**Figure 6.** Correlation between daily sap flow and influencing factors of *P. tabuliformis* and *R. pseudoacacia* with rainfall types II and III.

### 3.3. Hourly Sap Flow Characteristics within Three Rainfall Types

The hourly variation in sap flow for the three rainfall types is shown in Figure 7. The sap flow rate was higher after rainfall than before rainfall for the three rainfall types, and the difference in sap flow was the greatest for type II. The hourly sap flow of the two species showed significant diurnal variation, and the characteristics of sap flow before the rainfall showed a single peak curve only when type III decreased. The daytime sap flow of *P. tabulaeformis* and *R. pseudoacacia* accounted for 78.78–92.53% and 84.44–90.85% (from 5:00 to 19:00), respectively.

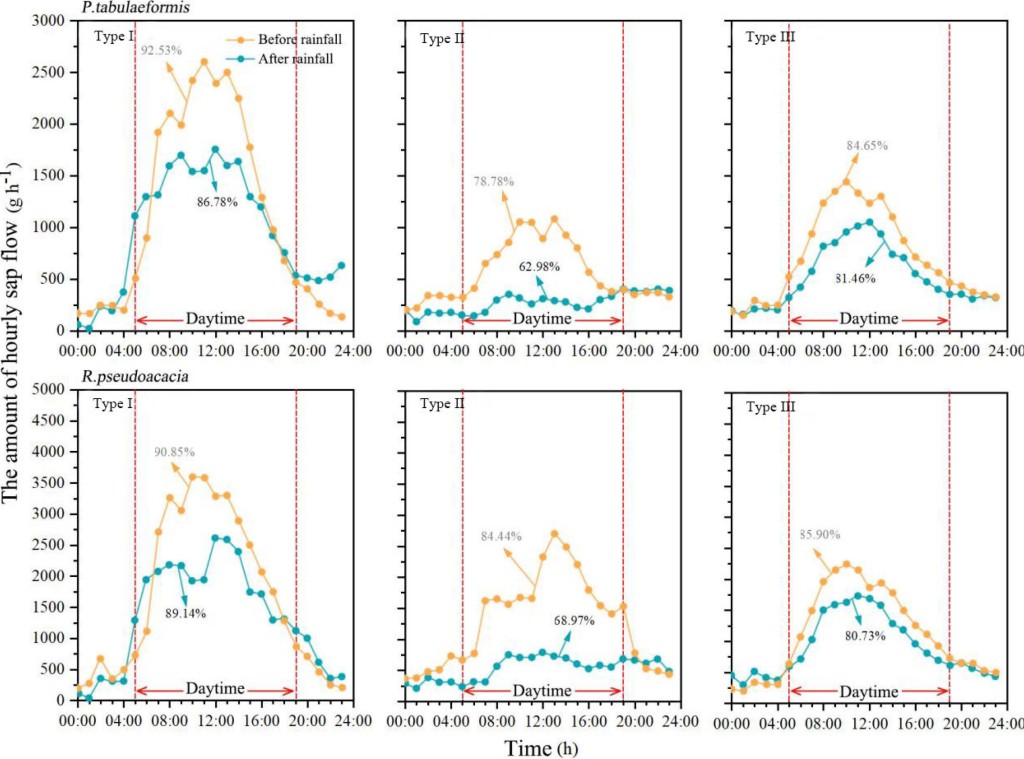

**Figure 7.** Variation in hourly sap flow of *P. tabuliformis* and *R. pseudoacacia* with three rainfall types (the values of mean diurnal curves were averaged among all rainfalls of this type and the % values were the daytime sap flow of two special trees (from 5:00 to 19:00) accounted for whole day (24 h)).

In the case of type I, the before rainfall sap flow of the two species was higher than that on the day of the rainfall for a short time at sunrise, and the "siesta" phenomenon of a transient decrease in sap flow occurred from 10:00 to 11:00. In the case of type III, the "siesta" phenomenon occurred at 12:00.

Three rainy days with similar sap flow characteristics were screened to illustrate the hourly sap flow characteristics of the two species (Figure 8). Compared with type III, the sap flow of *R. pseudoacacia* decreased, while *P. tabulaeformis* was relatively stable. The rapid recovery of sap flow after type I rainfall indicates that the recovery of sap flow has little relationship with the interval time after rainfall. In type II, the recovery of sap flow after rainfall was slow and the recovery range was low, and the sap flows of the two species were low, and *R. pseudoacacia* decreased more significantly than *P. tabulaeformis*. The decrease in the sap flow of *R. pseudoacacia* was more obvious than that of *P. tabulaeformis*. After rainfall, the nighttime sap flows of both species increased. In type III, the recovery of sap flow after rainfall was fast and the sap flow of *R. pseudoacacia* also increased after rainfall, and that of *P. tabulaeformis* was relatively stable. Therefore, the sap flow of *R. pseudoacacia* was more susceptible to rainfall than *P. tabulaeformis*, and the two species were different under the same rainfall types, with different rainfall distribution periods.

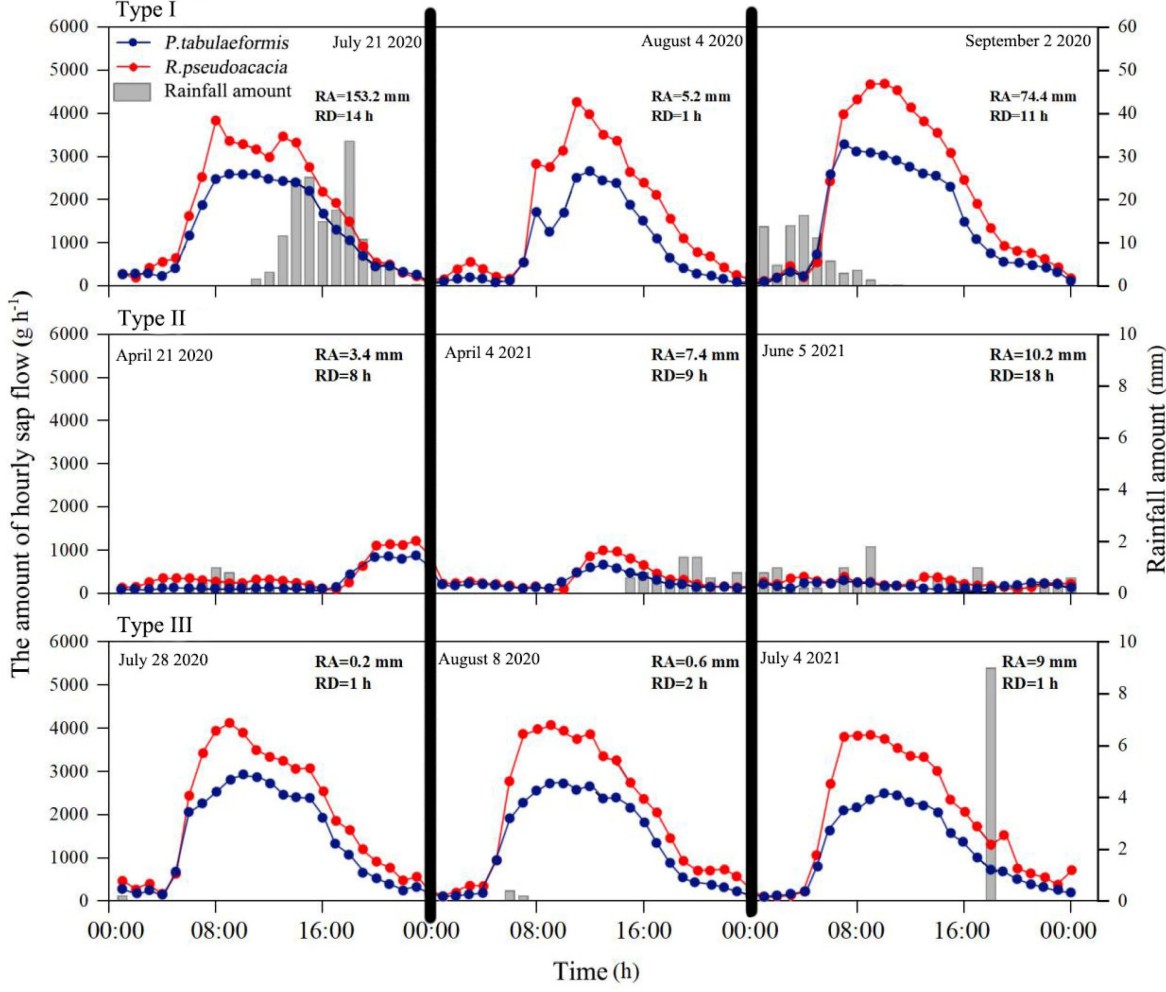

**Figure 8.** Variations in hourly sap flow of *P. tabuliformis* and *R. pseudoacacia* with three rainfall types (the three rainy days for type I were 21 July 2020, 4 August 2020 and 2 September 2020; the three rainy days for type II were 21 April 2020, 4 April 2021 and 5 June 5 2021; the three rainy days for type III were 28 July 2020 and 8 August 2020, and 4 July 2021.The time of all sap flow values was the corresponding time 24 h after rainy day).

## 4. Discussion

### 4.1. Response of Sap Flow Characteristics to Other Factors under Three Rainfall Types at the Daily Scale

The average daily solar radiation of type I rainfall was significantly higher than that of type III rainfall, while other meteorological factors (average daily temperature, relative humidity, and VPD) showed no significant difference among the three rainfall types. The highest daily sap flow values of the two species occurred with type I rainfall. At the daily scale, solar radiation and VPD were the main influencing factors of sap flow characteristics. The reason for this is that solar radiation changed the hydrothermal state of the soil, vegetation, and atmosphere, and then affected the distribution characteristics of the soil water content, soil water potential, and VPD. These factors have important effects on sap flow. Some studies have shown that a 74.37% variation in sap flow for broad-leaved trees in the Beijing mountain area of China was caused by solar radiation and temperature [35,38,41,42]. The daily sap flow rate of *R. pseudoacacia* was higher than that of *P. tabulaeformis*, which may be related to the difference in DBH between the two tree species and the difference in the physiological structures of conifer species. In this study, the tree height and DBH of *R. pseudoacacia* were higher than those of *P. tabulaeformis*. Previous studies have found that the xylem sapwood area affects the sap flow rate of the trunk, and the sap flow flux and sap flow rate of the large-diameter class were significantly higher than those of the small-diameter class. Broad-leaved tree species have more efficient water transport structures and larger leaves, and their sap flow characteristics are more responsive to meteorological factors [43].

The daily sap flows of *P. tabulaeformis* and *R. pseudoacacia* decreased significantly under type II rainfall, which may be because type II rainfall was characterized by a longer rainfall duration (more than 8 h). The long-term rainfall was usually accompanied by higher relative humidity, lower temperature, and solar radiation, which attenuated the transpiration pull and extended the drying time of the leaves. The increasing number of stomata and prolonged closure time reduced the transpiration rate of the trees [44]. In addition, the sap flow on the day after rainfall increased compared with the day before rainfall for the three rainfall types. Under type II, the average daily sap flows of *P. tabulaeformis* and *R. pseudoacacia* after rainfall increased by 85.2% and 98.7% compared with those before rainfall, a greater difference than that of types I and III rainfall. This result was contrary to other sap flow studies in sub-humid climate regions, where it has been found that the sap flow of trees before rainfall is higher than that after rainfall. The reason may be that the study samples were located in a relatively arid soil rock mountain area [45]. The lower soil water and higher transpiration force caused the trees to close their stomata to prevent water loss before rainfall, and the sap flow rate decreased. Rainfall replenished the soil water, and the increased water accelerated the transpiration rate of the trees. Changes in the soil water content led to greater nighttime sap flow on rainy days than on sunny days [42].

Type I rainfall is a special rainfall type in the study area, mainly composed of extremely heavy rainfall with a low frequency (three times). Types II and III rainfall were more common in the study area. There were differences in the response of the sap flow characteristics of *P. tabulaeformis* and *R. pseudoacacia* to meteorological factors under the two rainfall types. Under type II rainfall, the sap flow characteristics of *P. tabulaeformis* were positively correlated with temperature, solar radiation, and VPD, while those of *R. pseudoacacia* were only positively correlated with solar radiation. Under type III rainfall, the sap flow characteristics of the two species were also significantly positively correlated with soil moisture content and negatively correlated with relative humidity. This indicates that, compared with typical sunny days, on rainfall days, the influence of various meteorological factors on the sap flow characteristics of trees is restricted by the rainfall types. For type II rainfall, the rainfall duration was longer, and lower rainfall intensity usually leads to higher relative humidity all day. The sap flow characteristics of the trees were more affected by solar radiation. Under type III rainfall, the rainfall duration was shortest and intensity was the highest. The tree sap flow characteristics were more susceptible to the

influence of relative humidity and soil water status. Previous studies have shown that the root distribution of *P. tabulaeformis* and *R. pseudoacacia* was shallow, and in dry soil and rock mountains, high-intensity rainfall exerts a significant impact on the sap flow characteristics of shallow-rooted tree species by supplementing the soil water in shallow layers.

*4.2. Response of Sap Flow Characteristics to Other Factors under Three Rainfall Types at the Hourly Scale*

Under three rainfall types, the hourly sap flow amounts of *P. tabulaeformis* and *R. pseudoacacia* after rainfall were higher than before rainfall. Type I rainfall significantly promoted the sap flow of these two species. Its daytime sap flow rate was also higher, which may have been due to the high intensity of the rainfall supplementing shallow soil water, while the short-term changes in sap flow were more susceptible to environmental factors. VPD tended towards 0 during rainfall, resulting in the rate of sap flow tending towards 0. As for the flow under type I rainfall, the sap flow also increased from 0 earlier, which may be related to the higher solar radiation possessed by type I rainfall, or because the higher solar radiation possessed by the type I rainfall start time of the sap flow lagged behind the solar radiation. The time lag between the trunk flow and canopy transpiration was less, almost negligible.

Both types I and III rainfall had the phenomena of a midday depression, mainly because of the stronger solar radiation of these two types. The root water absorption from the soil was less than the transpiration consumption. The strongest solar radiation plants closed their leaf stomata to prevent water loss, which was consistent with the change in solar radiation. As for type I at 11:00 and 10:00, the flow fluctuation was greater, the midday depression on rainfall days occurred at 12:00, and the rainfall flow was a single peak curve with no "midday depression" phenomenon. The characteristics of type I rainfall flow before and after showed the biggest difference, and the daytime flow before rainfall was significantly lower than that with other rainfall types. The flow fluctuation on rainfall days was larger, which may be because the class rainfall lasted the longest, and the change in environmental factors was relatively complex. Plants adapt to environmental changes through their water capacity adjustment. Research has shown that unsteady flow characteristics were mainly the result of their own regulation. Contrasting the three types of rainfall represents the change in flow characteristics. For two tree species, type II rainfall had decreased. The sap flow rate in the rainfall period was the result of transpiration tension and soil water supplementation. The rainfall occurred in the daytime, although two species of sap flow at night have significantly improved, and night hydrating may occur.

*4.3. The Difference in Sap Flow Characteristics between Conifer and Broad-Leaved Species under Three Rainfall Types*

The effects of different types of rainfall species on the rate of increase in sap flow were different, with different influences on the soil moisture status and tree transpiration rate. Rainfall significantly promoted an increase in sap flow rate, promoting the sap flow effect of *R. pseudoacacia* over *P. tabulaeformis*. The type I rainfall promotion effect was small, but still greater than *R. pseudoacacia*, which may suggest that different species of trees' sap flow characteristics respond differently to the three rainfall types.

The study shows that the flow characteristics of broad-leaf species are more susceptible to environmental factors, but this study found that, under different rainfall types, the sap flow was higher than *P. tabulaeformis*. However, *P. tabulaeformis* was more sensitive to meteorological factors, which may be related to the responses of the two species to different rainfall types, and the transpiration rate was more susceptible to environmental factors other than solar radiation.

The daytime sap flow decreased after type I rainfall, and the suppression of the *R. pseudoacacia* sap flow rate was more obvious, indicating that the *R. pseudoacacia* flow rate on rainfall days was more likely to receive inhibition. The sap flow of *R. pseudoacacia* rebounded significantly after rainfall, while the sap flow of *P. tabulaeformis* was relatively stable under the three types of rainfall. This may be related to the physiological structures

of *R. pseudoacacia* and *P. tabulaeformis*. The leaf structure can affect the plant's interception of precipitation, thus affecting the time consumed by leaf drying and stomatal opening and closing [46].

## 5. Conclusions

The daily sap flow of *R.pseudoacacia* and *P. tabulaeformis* was significantly reduced under the type II rainfall ($p < 0.05$). The sap flow characteristics of *R. pseudoacacia* were positively correlated with solar radiation ($p < 0.05$), while those of *P. tabulaeformis* were significantly positively correlated with air temperature, solar radiation, and VPD ($p < 0.01$); the sap flow characteristics of *R. pseudoacacia* and *P. tabulaeformis* were significantly positively correlated with air temperature, solar radiation, VPD, and soil water content ($p < 0.01$), and were negatively correlated with the relative humidity ($p < 0.05$).

The sap flows of both *R. pseudoacacia* and *P. tabulaeformis* after rainfall were higher than before rainfall at the daily scale. Type I rainfall improved the sap flow of the two tree species, and the proportion of the daytime sap flow was higher, whilst the sap flow of *R. pseudoacacia* and *P. tabulaeformis* showed a "midday depression" phenomenon after rainfall types I and III. For type I, the "midday depression" phenomenon after rainfall occurred an hour earlier than that before rainfall type III. The flow of type III showed a unimodal curve, while the type II rainfall significantly inhibited the daytime flow of *R. pseudoacacia* and *P. tabulaeformis*. The flow rate rebounded at night after rainfall.

The daily sap flow of *R. pseudoacacia* was more easily affected by the type of rainfall, while *P. tabulaeformis* was more susceptible to types I and III. *P. tabulaeformis* was more sensitive to changes in meteorological factors, and the sap flow rate of *R. pseudoacacia* was more easily inhibited on rainy days. *P. tabulaeformis* was relatively stable.

The main factors affecting the sap flow characteristics differed under different time scales and rainfall types. When studying tree trunk sap flow characteristics in arid environments, the rainfall types should be considered in addition to conventional meteorological factors to clarify the influence of various rainfall parameters on the sap flowing from the tree trunk. The results further reveal the hydrological processes and some physiological hydrological driving mechanisms of forest ecosystems in northern China, and provide a technical reference for the accurate estimation of forest ecosystem water carrying capacity and sustainable forest management under seasonal drought.

**Author Contributions:** Y.C. conducted data curation, methodology and writing—original draft. Y.W. conducted data curation, formal analysis, software and writing—original draft. N.Z., C.N. and Y.B. conceived investigation and project administration. J.J. conducted methodology, writing—original draft and writing—review and editing. All authors have read and agreed to the published version of the manuscript.

**Funding:** This research was funded by Natural and Science Basic Research Program of Shaanxi Province (2022JQ-295), Hunan Province Natural Science Foundation (2023JJ31003), Changsha Natural Science Foundation Project (KQ2208413), and Hunan Province key research and development project (2023SK2055). The authors gratefully acknowledge all financial support for this study.

**Data Availability Statement:** All relevant data are within the manuscript.

**Conflicts of Interest:** Yongxiang Cao, Naichang Zhang, Chendong Ning, and Yu Bai were employed by the company of Power China Northwest Engineering Corporation Limited. The remaining authors declare that the research was conducted in the absence of any commercial or financial relationships that could be construed as potential conflicts of interest.

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
