# Peer review of "Response of Sap Flow Trends of Conifer and Broad-Leaved Trees to Rainfall Types in Sub-Humid Climate Region of China"

_water, doi:10.3390/w16010095_

Round 1

Reviewer 1 Report

Comments and Suggestions for Authors

The article “Response of Sap Flow Trends of Conifer and Broadleaved Trees to Rainfall Types in Sub-Humid Climate Region of China” is devoted to a current topic: assessment of sap flow as the most important physiological process of water transport of trees.

Mentions of formulas in the text should be numbered: lines 119, 126, 141.

The discussion of the results obtained, and the number of literature sources should be expanded. Now there are only 29 sources.

Author Response

Dear Reviewers:

Thank you for your letter and for the comments concerning our manuscript entitled “Response of Sap Flow Trends of Conifer and Broad-leaved Trees to Rainfall Types in Sub-Humid Climate Region of China”. I am sorry that I wrote back after such a long time. Those comments were all valuable and very helpful for revising and improving our paper, as well as the important guiding significance to our researches. We have studied comments carefully and have made correction which we hope meet with approval. At the same time, we revised a lot of grammatical and lexical errors in the whole paper. Revised portion are marked in the paper. The main corrections in the paper and the responds to the reviewer’s comments are as flowing:

Reviewer #:    

The article“Response of Sap Flow Trends of Conifer and Broadleaved Trees to Rainfall Types in Sub-Humid Climate Region of China” is devoted to a current topic: assessment of sap flow as the most important physiological process of water transport of trees.

Comment 1:Mentions of formulas in the text should be numbered: lines 119, 126, 141.

Response 1:All formulas in the manuscript have been numbered.

Comment 2:The discussion of the results obtained, and the number of literature sources should be expanded. Now there are only 29 sources.

Response 2:In discussion part, we have added further discussion on the research results and the the number of literature to make this part more complete and reasonable.

Reviewer 2 Report

Comments and Suggestions for Authors

The paper is about the effect of rainfall patterns on the sap flow. Taking into account not only the rain amount, but also how it is distributed in time his is an important topic. However the paper must be dramatically improved.

First, English is rather problematic. Singular and plural, present and past are often mixed up, some sentences are unclear. I corrected some errors in first 2-3 pages (see attachment), but much more corrections needed.

In Abstract it should be mentioned what are rain types I-III

In methods you must indicate the period of measurements.

How individual rainfall events were separated from each other? Did you assume that if it was no rain for 15 minutes and then rain restarted, these are two different rain events? Or if it was a day without rain?

It is not clear what means sap flow for rain event. Do you mean sap flow directly during the rain? If yes, it makes not much sense, as when leaves are wet, transpiration anyway should decrease. If you mean AFTER the rain, then you must say it clearly and also mention how long period after the rain event you attribute to this event.

As rain affects sap flow mainly via soil water content (SWC) and in a smaller extend via VPD, it would be good to show, how SWC at different layers reacts to different rain types, not only immediately, but also in the seasonal time scale. Generally, the effect of rain on sap flow should be considered in connection with SWC, VPD and radiation, not only during rain event, but also between rain events.

On Fig. 8 there is no considerable variation of sap flow except of full stop of it in last day of type II (when it was very low the whole 3-days period). To judge about the differences between rain types in this figure it would be good to see also SWC and VPD.

Many authors noticed that Granier type sap flow sensors can require species-specific calibration. Do you know if anybody did it already for your species? Also, as Granier sensors measure in one depth (which one in your case?) and sap flow radial profile is very nonhomogeneous, a correction for radial profile can be necessary (see, e.g., Paudel et al., 2013, doi:10.1093/treephys/tpt070).

When describing diurnal pattern of sap flow, it makes sense to compare rainy and dry days, this description for only rainy days makes not much sense, as sap flow pattern can depend on the rain timing.

As sap flow reaction is buffered relatively to transpiration, it would be also good to analyze the time lags between sap flow and driving variables (including rain pattern)

Other comments are in the attachment

Comments on the Quality of English Language

English is rather problematic. Singular and plural, present and past are often mixed up, some sentences are unclear. I corrected some errors in first 2-3 pages (see attachment), but much more corrections needed.

Author Response

Dear Reviewers:

Thank you for your letter and for the comments concerning our manuscript entitled “Response of Sap Flow Trends of Conifer and Broad-leaved Trees to Rainfall Types in Sub-Humid Climate Region of China”. I am sorry that I wrote back after such a long time. Those comments were all valuable and very helpful for revising and improving our paper, as well as the important guiding significance to our researches. We have studied comments carefully and have made correction which we hope meet with approval. At the same time, we revised a lot of grammatical and lexical errors in the whole paper. Revised portion are marked in the paper. The main corrections in the paper and the responds to the reviewer’s comments are as flowing:

Reviewer:    

The paper is about the effect of rainfall patterns on the sap flow. Taking into account not only the rain amount, but also how it is distributed in time his is an important topic. However the paper must be dramatically improved.

Comment 1:First, English is rather problematic. Singular and plural, present and past are often mixed up, some sentences are unclear. I corrected some errors in first 2-3 pages (see attachment), but much more corrections needed.

Response 1:Thank you very much for your valuable comments on the manuscript. The whole manuscript has been carefully revised considering the language and grammar problems, including the sentences that were convoluted and hard to follow according to the attachment comments. At the same time, the manuscript have been reviewed by an experienced English-speaking colleague.

Comment 2:In Abstract it should be mentioned what are rain types I-III.

Response 2:The definition of three rainfall types have been explained in abstract in L18-19.

Comment 3:In methods you must indicate the period of measurements.

Response 3:The period of measurements have lasted a total of 316 days (from April 21 to September 28, 2020 and from April 4 to September 5, 2021) and have been explained in methods in 2.4. Cluster Analysis of Rainfall Events.

Comment 4:How individual rainfall events were separated from each other? Did you assume that if it was no rain for 15 minutes and then rain restarted, these are two different rain events? Or if it was a day without rain?

Response 4:I am very sorry that I did not explain clearly the problem of the division of rainfall events. Although there is no clear regulation on the interval between two rainfall events, according to some rainfall events research, the manuscript define the interval between two rainfall events as more than 24 hours. The parts have been explained in methods in 2.2. Sap Flow Measurements.

Comment 5:It is not clear what means sap flow for rain event. Do you mean sap flow directly during the rain? If yes, it makes not much sense, as when leaves are wet, transpiration anyway should decrease. If you mean AFTER the rain, then you must say it clearly and also mention how long period after the rain event you attribute to this event.

Response 5:Thank you for your question. The manuscript researches the influence of different rainfall events on sap flow, that is, the difference analysis of sap flow before rainfall and after rainfall. According the manuscript define the interval between two rainfall events as more than 24 hours, the sap flow before rainfall (or pre-rainfall in Figure 5) refers to the sap flow within 24 hours before one rainfall event begins, and sap flow after rainfall (or rainfall in Figure 5) refers to the sap flow within 24 hours before one rainfall event stops. The parts have been explained in methods in 2.2. Sap Flow Measurements.

Comment 6:As rain affects sap flow mainly via soil water content (SWC) and in a smaller extend via VPD, it would be good to show, how SWC at different layers reacts to different rain types, not only immediately, but also in the seasonal time scale. Generally, the effect of rain on sap flow should be considered in connection with SWC, VPD and radiation, not only during rain event, but also between rain events.

Response 6:Thank you for your good suggestion. The characteristics of soil water content (different layers) and VDP between rainfall events can indeed better reflect the characteristic of different rainfall events and effect of sap flow. In our next study, we will focus on the changes of hydrological process of soil-vegetation-atmosphere continuum caused by different rainfall events.

Comment 7:On Fig. 8 there is no considerable variation of sap flow except of full stop of it in last day of type II (when it was very low the whole 3-days period). To judge about the differences between rain types in this figure it would be good to see also SWC and VPD.

Response 7:Thank you for your good suggestion. Figure 8 was the three rainy days with similar sap flow characteristics, not three consecutive days. The three rainy days of type I were July 21, August 4 and September 2 in 2020, respectively; Type II were April 21 in 2020, April 4 and June 5 in 2021, respectively; Type III were July 28 and August 8 in 2020, July 4 in 2021, respectively. The parts have been explained in Figure 8. In addition, the characteristic of type II is the longest rainfall duration that cause soil water content and water vapor saturation were relatively constant, resulting in low sap flow on the whole period, but there were also fluctuations. The main consideration was the influence of the physiological and hydrological processes of trees (such as the rehydrate effect of trees at night, etc.). These explain was written in the last part of the discussion.

Comment 8:Many authors noticed that Granier type sap flow sensors can require species-specific calibration. Do you know if anybody did it already for your species? Also, as Granier sensors measure in one depth (which one in your case?) and sap flow radial profile is very nonhomogeneous, a correction for radial profile can be necessary (see, e.g., Paudel et al., 2013, doi:10.1093/treephys/tpt070).

Response 8:Thank you for your comments. As for the calibration of Granier sensors, we have already considered radial calibration and tested the accuracy of the flow when installing the Granier sensors. Two kinds of probes (20mm and 30mm) were selected according to the xylem catheter in radial position to ensure the accurate position of the probe.

Comment 9:When describing diurnal pattern of sap flow, it makes sense to compare rainy and dry days, this description for only rainy days makes not much sense, as sap flow pattern can depend on the rain timing.

Response 9:Thank you for your suggestion. Current research of sap flow focus on the comparison of sap flow characteristics in typical sunny and rainy days, or dry and rainy seasons and the purpose is to provide data support for the study of vegetation transpiration and hydrological process. Our study aimed to investigate the effects of different rainfall events on sap flow for the purpose of providing water use efficiency, which is a supplement to the research on vegetation transpiration and hydrological processes.

Comment 10:As sap flow reaction is buffered relatively to transpiration, it would be also good to analyze the time lags between sap flow and driving variables (including rain pattern).

Response 10:Thank you for your good suggestion. In the next study, we will carry out the time lags between sap flow and rain pattern variables research. Thanks again for your help.

Comment 11:Other comments are in the attachment.

Response 11:Thank you very much for your detailed and comprehensive comments. Those comments were all valuable and very helpful for revising and improving our paper, as well as the important guiding significance to our researches. We have revised and explained each suggestion in the corresponding position according to all the modification suggestions you put forward. We will send you the revised version for your review.

Reviewer 3 Report

Comments and Suggestions for Authors

Comments: Response of Sap Flow Trends of Conifer and Broadleaved Trees to Rainfall Types in Sub-Humid Climate Region of China

I have read the manuscript with great interest and find it very useful and relevant under Sub-Humid regions. The abstract is well written giving brief highlights of the findings. The study validly adopts K-means clustering analysis to divide the rainfall in the study period into three rainfall types based on rainfall amount and intensity to compare the differences in response of sap flow trends and influencing factors of Pinus tabulaeformis and Robinia pseudoacacia under different rainfall types. There are, however, frequent grammatical and flow concerns even in the abstract (eg. compared, differences response etc). 

Introduction: It is disjointed and lacks fluency, and pith and fails to provide a proper background to the topic. It is mainly a type of literature survey. The obectives, as the stand currently, are not clear, concise and have some grammar issues upon first read: 'The main objectives of this study were:(i) explore(d)/ exploring the response difference of sap flow characteristics to different rainfall types; and (ii) (further) predicted? (predicting) sap flow trends in rainfall(y?) days, clarifying the effects of precipitation, rainfall duration and rainfall intensity on sap flow trends' whether trend is not a sap flow character and if so, why it is designated as a separate objective? The introduction is also devoid of highlighting the research gap or any other contribution to science. 

Material/MEthod: Please consider using past sentence about the study area description. 'The basic information of the plot and tree was (is?)shown in Table 1.

Results and Discussion: The results are intuitive and interesting however very few relevant studies have been referred in these sections to compare or contrast or validate the current findings. 

Conclusion: The conclusion just gives a summary highlight of the finding with no proper implications and study perspective for policy and research.

Comments on the Quality of English Language

It needs some work.

Author Response

Dear Reviewers:

Thank you for your letter and for the comments concerning our manuscript entitled “Response of Sap Flow Trends of Conifer and Broad-leaved Trees to Rainfall Types in Sub-Humid Climate Region of China”. I am sorry that I wrote back after such a long time. Those comments were all valuable and very helpful for revising and improving our paper, as well as the important guiding significance to our researches. We have studied comments carefully and have made correction which we hope meet with approval. At the same time, we revised a lot of grammatical and lexical errors in the whole paper. Revised portion are marked in the paper. The main corrections in the paper and the responds to the reviewer’s comments are as flowing:

Reviewer:    

I have read the manuscript with great interest and find it very useful and relevant under Sub-Humid regions. The abstract is well written giving brief highlights of the findings. The study validly adopts K-means clustering analysis to divide the rainfall in the study period into three rainfall types based on rainfall amount and intensity to compare the differences in response of sap flow trends and influencing factors of Pinus tabulaeformis and Robinia pseudoacacia under different rainfall types. There are, however, frequent grammatical and flow concerns even in the abstract (eg. compared, differences response etc).

Comment 1:Introduction: It is disjointed and lacks fluency, and pith and fails to provide a proper background to the topic. It is mainly a type of literature survey. The obectives, as the stand currently, are not clear, concise and have some grammar issues upon first read: 'The main objectives of this study were:(i) explore(d)/ exploring the response difference of sap flow characteristics to different rainfall types; and (ii) (further) predicted? (predicting) sap flow trends in rainfall(y?) days, clarifying the effects of precipitation, rainfall duration and rainfall intensity on sap flow trends' whether trend is not a sap flow character and if so, why it is designated as a separate objective? The introduction is also devoid of highlighting the research gap or any other contribution to science.

Response 1:Thank you for your good suggestion. In the discussion section, we have added relevant research literature to explain the research background and scientific question. At the same time, we have revised the second research objectives and added the research significance.

Comment 2:Material/MEthod: Please consider using past sentence about the study area description. 'The basic information of the plot and tree was (is?)shown in Table 1.

Response 2:Thank you for your suggestion. The whole manuscript has been carefully revised considering the language and grammar problems, including the sentences that were convoluted and hard to follow according to the attachment comments. At the same time, the manuscript have been reviewed by an experienced English-speaking colleague. 

Comment 3:Results and Discussion: The results are intuitive and interesting however very few relevant studies have been referred in these sections to compare or contrast or validate the current findings.

Response 3:Thank you for your suggestion. In the discussion section, we have added relevant research literature to support and explain the phenomena and results in our research. Thanks again for your help.

Comment 4:Conclusion: The conclusion just gives a summary highlight of the finding with no proper implications and study perspective for policy and research.

Response 4:Thank you for your good suggestion. The manuscript purpose is to provide data support for the study of vegetation transpiration and hydrological process. Our study aims to investigate the effects of different rainfall events on sap flow for the purpose of providing water use efficiency, which is a supplement to the research on vegetation transpiration and hydrological process. At the same time, the results further reveal the hydrological processes and some physiological hydrological driving mechanisms of forest ecosystems in North China, and provide technical reference for accurate estimation of forest ecosystem water carrying capacity and sustainable forest management under seasonal drought. These explain was written in introduction and the last part of the conclusion. 

Round 2

Reviewer 2 Report

Comments and Suggestions for Authors

The authors strongly improved the paper.

I have not much notes remained. The main one is about Fig.8.

(1) In all 9 rain events on this figure there is no visible reaction of sap flow on the rain. Is it common situation? Generally one can expect that during the rain transpiration must decrease (because of wet leaves, low VPD and cloudiness). If sap flow did not react, it means its considerable time lag after the transpiration. Consider analyzing and discuss these issues.

(2) Sap flow for all 3 rains of Type II is much lower than at types I and III. Is it typical for type II or it is occasional result of particular 34 days selection (when comparing with Fig. 4 it looks like the later)?

(3) 3 days on the fig. look like continuous, in particular sap flow curves are connected, 00:00 moments look like the borders between two days etc. But in fact these are not consequent days. So, I suggest to make gaps between days on the fig.

(4) The discussion about diurnal dynamics contradicts Fig.8 (see attachment)

As I understood from your answer to my previous comments, “under rainfall” usually should mean “during 24 h after rainfall”. But e.g., Fig.8 represents data clearly during rainfall. So, make it clear, when you mention during and when after rainfall.

In Response 8 you wrote: “As for the calibration of Granier sensors, we have already considered radial calibration and tested the accuracy of the flow when installing the Granier sensors. Two kinds of probes (20mm and 30mm) were selected according to the xylem catheter in radial position to ensure the accurate position of the probe.” It would be good to mention it in the Methods

Sometimes different sentences contradict to each other or to figures (see attachment)

Comments on the Quality of English Language

English was strongly improved. Some notes are in the attachment

Author Response

Reviewer #:    

The authors strongly improved the paper. I have not much notes remained. The main one is about Fig.8.

Comment 1:In all 9 rain events on this figure there is no visible reaction of sap flow on the rain. Is it common situation? Generally one can expect that during the rain transpiration must decrease (because of wet leaves, low VPD and cloudiness). If sap flow did not react, it means its considerable time lag after the transpiration. Consider analyzing and discuss these issues.

Response 1:Thank you for your good suggestion. The time of all sap flow values was the corresponding time 24 hours after rainy day. Some rain events were no visible reaction of sap flow after rainy day and the other events were visible reaction of sap flow after rainy day.

Comment 2:Sap flow for all 3 rains of Type II is much lower than at types I and III. Is it typical for type II or it is occasional result of particular 34 days selection (when comparing with Fig. 4 it looks like the later)?

Response 2:Thank you for your question. This was a particular result only from the sap flow value in Figure 8, which reflects the small change of sap flow value after type II rainfall, rather than the absolute value of sap flow.

Comment 3:3 days on the fig. look like continuous, in particular sap flow curves are connected, 00:00 moments look like the borders between two days etc. But in fact these are not consequent days. So, I suggest to make gaps between days on the fig.

Response 3:Thank you for your good suggestion. We redrew Figure 8, adding spacing lines and adding the date of rainfall on each typical day.

Comment 4:The discussion about diurnal dynamics contradicts Fig.8 (see attachment)

Response 4:Thank you for your question. L314 you mentioned “It does not follow from th Fig.8. Diurnal dynamics in all 3 days is very similar, although the rain in these days was different time (in the afternoon, next midnight and in the morning). As you did not show diurnal dynamics in the days without rain, it is hard to compare”. This result really cannot be compared from Fig.8. We didn't make that part clear and rephrase that “The rapid recovery of sap flow after type I rainfall indicates that the recovery of sap flow has little relationship with the interval time after rainfall. In type II, the recovery of sap flow after rainfall was slow and the recovery range was low”. At the same time, L406 you mentioned “It contradicts to Fig.8, where sap flow is far from 0 until the night”. Sorry for our lack of clarity. What we want to express was that the change of sap flow rate tends to 0, rather than the sap flow value tends to 0. We have modified it in this part.  

Comment 5:As I understood from your answer to my previous comments, “under rainfall” usually should mean “during 24 h after rainfall”. But e.g., Fig.8 represents data clearly during rainfall. So, make it clear, when you mention during and when after rainfall.

Response 5:Thank you for your question. In Fig.8, the time of all sap flow values was the corresponding time 24 hours after rainy day. We added a description in the title of Figure 8.

Comment 6:In Response 8 you wrote: “As for the calibration of Granier sensors, we have already considered radial calibration and tested the accuracy of the flow when installing the Granier sensors. Two kinds of probes (20mm and 30mm) were selected according to the xylem catheter in radial position to ensure the accurate position of the probe.” It would be good to mention it in the Methods.

Response 6:Thank you for your good suggestion.The parts have been mentioned in methods in 2.2. Sap Flow Measurements.

Comment 7:Sometimes different sentences contradict to each other or to figures (see attachment).

Response 7:Thank you very much for your detailed and comprehensive comments again. We have revised and explained each suggestion in the corresponding position according to all the modification suggestions you put forward.

Reviewer 3 Report

Comments and Suggestions for Authors

The authors have aptly addressed my comments and improved the article significantly.

Author Response

Comment ï¼šThe authors have aptly addressed my comments and improved the article significantly.

Response ï¼šThank you again for your recognition of our work, and we will strive to publish the article as soon as possible.